# CoF-CoT: Enhancing Large Language Models with Coarse-to-Fine Chain-of-Thought Prompting for Multi-domain NLU Tasks

**Hoang H. Nguyen[1], Ye Liu[2], Chenwei Zhang[3], Tao Zhang[1], Philip S. Yu[1]**

[1] Department of Computer Science, University of Illinois at Chicago, Chicago, IL, USA
[2] Salesforce Research, Palo Alto, CA, USA
[3] Amazon, Seattle, WA, USA

`{hnguy7,tzhang90,psyu}@uic.edu`, `yeliu@salesforce.com`, `cwzhang@amazon.com`

## Abstract

While Chain-of-Thought prompting is popular in reasoning tasks, its application to Large Language Models (LLMs) in Natural Language Understanding (NLU) is under-explored. Motivated by multi-step reasoning of LLMs, we propose Coarse-to-Fine Chain-of-Thought (CoF-CoT) approach that breaks down NLU tasks into multiple reasoning steps where LLMs can learn to acquire and leverage essential concepts to solve tasks from different granularities. Moreover, we propose leveraging semantic-based Abstract Meaning Representation (AMR) structured knowledge as an intermediate step to capture the nuances and diverse structures of utterances, and to understand connections between their varying levels of granularity. Our proposed approach is demonstrated effective in assisting the LLMs adapt to the multi-grained NLU tasks under both zero-shot and few-shot multi-domain settings [1].

## 1 Introduction

Natural Language Understanding (NLU) of Dialogue systems encompasses tasks from different granularities. Specifically, while intent detection requires understanding of coarse-grained sentence-level semantics, slot filling requires fine-grained token-level understanding. Moreover, Semantic Parsing entails the comprehension of connections between both token-level and sentence-level tasks.

Large Language Models (LLMs) possess logical reasoning capability and have yielded exceptional performance (Zoph et al., 2022; Zhao et al., 2023b). However, they remain mostly restricted to reasoning tasks. On the other hand, mutli-step reasoning can take place when solving multiple interconnected tasks in a sequential order. In practical NLU systems, as coarse-grained tasks are less challenging, they can be solved first before proceeding to fine-grained tasks. Therefore, the coarse-grained tasks' outcomes can provide

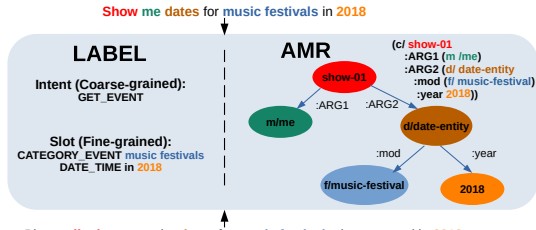

Figure 1: Illustration of **Abstract Meaning Representation (AMR)** of two structurally different but semantically similar utterances with the same fine-grained and coarse-grained labels. Each colored node represents an AMR concept matching the colored word or phrase existent in the corresponding utterances.

valuable guidance towards subsequent fine-grained tasks, allowing for deeper semantic understanding of diverse utterances across different domains within NLU systems (Firdaus et al., 2019; Weld et al., 2022; Nguyen et al., 2023a). For instance, consider the utterance *"Remind John the meeting time at 8am"* under **reminder** domain, recognizing **GET_REMINDER_DATE_TIME** intent is crucial for correctly understanding the existence of **PERSON_REMINDED** slot type rather than **CONTACT** or **ATTENDEE** slot type.

Chain-of-Thought (CoT) (Wei et al., 2022) provides an intuitive approach to elicit multi-step reasoning from LLMs automatically. However, there remain two major challenges with the current CoT approach: (1) LLMs entirely rely on their uncontrollable pre-trained knowledge to generate step-by-step reasoning and could result in unexpected hallucinations (Yao et al., 2022; Zhao et al., 2023a), (2) Additional beneficial structured knowledge cannot be injected into LLMs via the current CoT.

On the other hand, structured representation demonstrates the effectiveness in enhancing the capability of Pre-trained Language Models (PLMs) (Xu et al., 2021; Bai et al., 2021; Shou et al., 2022). In Dialogue systems, the dependencies among different dialogue elements together with the existent diversely structured utterances necessitate the inte-

---

[1] https://github.com/nhhoang96/CoF-CoT

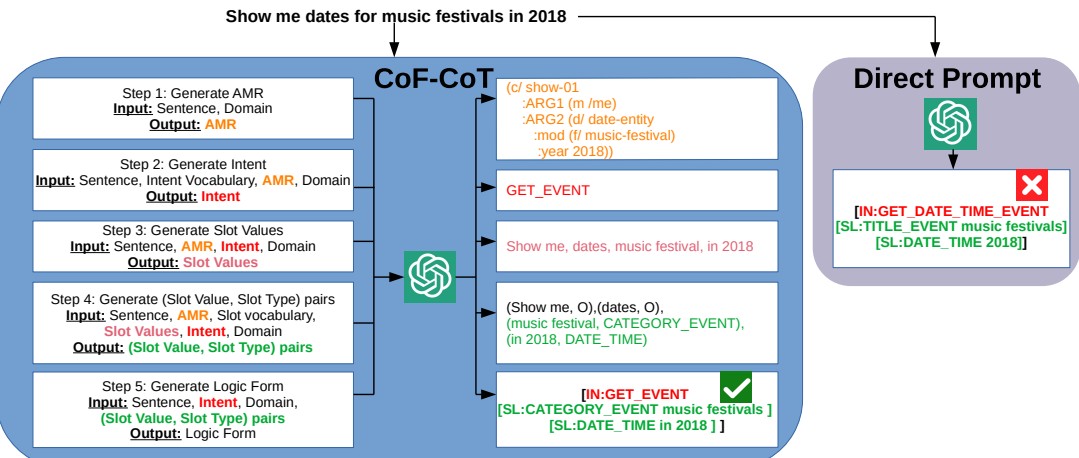

Figure 2: Illustration of **CoF-CoT** and its counterpart **Direct Prompt** approach. The left side illustrates the proposed CoF-CoT. The right side illustrates the naive Direct Prompt approach. **Red** and **Green** represent **sentence-level** and **token-level** annotations captured in the Logic Form respectively. For CoF-CoT, the prompt at each step starting from Step 2 is conditioned on the relevant output predicted from the previous step(s).

gration of additional structured representation. For instance, as observed in Figure 1, by leveraging Abstract Meaning Representation (AMR) (Banarescu et al., 2013), it is possible to map multiple semantically similar but structurally different utterances with similar coarse-grained and fine-grained labels into the same structured representation, allowing for effective extraction of intents, slots, and their interconnections within the Dialogue systems.

In our work, we explore the capability of LLMs in NLU tasks from various granularities, namely multi-grained NLU tasks. Motivated by CoT, we propose an adaptation of CoT in solving multi-grained NLU tasks with an integration of structured knowledge from AMR Graph. Our contribution can be summarized as follows:

• To the best of our knowledge, we conduct the first preliminary study of LLMs' capability in multi-grained NLU tasks of the Dialogue systems.

• We propose leveraging a CoT-based approach to solve multi-grained NLU tasks in a coarse-to-fine-grained sequential reasoning order.

• We propose integrating structured knowledge represented via AMR Graph in the multi-step reasoning to capture the shared semantics across diverse utterances within the Dialogue systems.

## 2 Related Work

**Chain-of-Thought (CoT)** CoT (Wei et al., 2022) proposes leveraging intermediate steps to extracts logical reasoning of LLMs and succeeds in various reasoning tasks. Wang et al. (2022) enhances CoT by selecting the most consistent output answers via majority voting. Additionally, Fu et al. (2022) argues majority consistency voting works

best among the most complex outputs. They propose complexity metrics and leverage them to select demonstration samples and decoding outputs. Unlike previous CoT approaches, we leverage CoT to solve multi-grained NLU tasks.

**Structured Representation** Structured Representation has been widely incorporated in language models to further enhance the capability across various NLP tasks (Bugliarello and Okazaki, 2020; Zhang et al., 2020). Structured representation can be either in the syntax-based structure (Bai et al., 2021; Xu et al., 2021) such as Dependency Parsing (DP) Graph, Constituency Parsing (CP) Graph or semantic-based structure (Shou et al., 2022) such as Abstract Meaning Representation (AMR) Graph (Banarescu et al., 2013). Unlike previous works, we aim at leveraging structured representation as an intermediate step in the multi-step reasoning approach to extract essential concepts from diverse utterances in the multi-domain Dialogue systems.

## 3 Proposed Framework

In this section, we introduce our proposed Coarse-to-Fine Chain-of-thought (CoF-CoT) approach for NLU tasks as depicted in Figure 2. Specifically, we propose a breakdown of multi-grained NLU tasks into 5 sequential steps from coarse-grained to fine-grained tasks. At each step, LLMs leverage the information from the previous steps as a guidance towards the current predictions. As domain name could provide guidance to NLU tasks (Xie et al., 2022; Zhou et al., 2023), at each step, we condition the *domain name* of the given utterance in the input prompt. The model's output is in the format of Logic Form (Kamath and Das) which encapsulates

Table 1: Experimental results on MTOP and MASSIVE under zero-shot and few-shot multi-domain settings.

| | MTOP | | | | | | | |
| --- | --- | --- | --- | --- | --- | --- | --- | --- |
| **Model** | **Zero-shot** | | | | **Few-shot** | | | |
| | NLU | | Semantic Parsing | | NLU | | Semantic Parsing | |
| | Intent Acc | Slot F1 | Frame Acc | Exact Match | Intent Acc | Slot F1 | Frame Acc | Exact Match |
| Direct Prompt | 31.50 ± 1.80 | 21.84 ± 2.83 | 8.33 ± 1.44 | 6.00 ± 1.32 | 51.33 ± 3.40 | 28.35 ± 3.24 | 11.00 ± 1.80 | 8.33 ± 1.00 |
| CoT | 31.83 ± 2.02 | 22.40 ± 1.61 | 8.67 ± 0.35 | 6.33 ± 1.04 | 47.67 ± 5.20 | 28.46 ± 3.10 | 11.83 ± 1.53 | 8.50 ± 1.04 |
| SC-CoT | 32.50 ± 1.89 | 22.71 ± 2.44 | 10.05 ± 0.87 | 6.83 ± 0.76 | 53.50 ± 3.04 | 29.53 ± 1.99 | 12.50 ± 1.80 | 9.00 ± 0.87 |
| ComplexCoT | 32.67 ± 2.00 | 22.86 ± 3.17 | 10.83 ± 0.29 | 7.16 ± 0.58 | 48.83 ± 2.47 | 29.21 ± 2.65 | 13.17 ± 0.58 | 8.83 ± 2.89 |
| Least-to-Most | 45.67 ± 0.58 | 21.84 ± 1.91 | **14.50 ± 0.50** | 8.00 ± 0.50 | 49.83 ± 4.54 | 27.28 ± 2.41 | 16.00 ± 0.50 | 8.83 ± 0.76 |
| Plan-and-Solve | 45.00 ± 4.00 | 22.45 ± 2.28 | 9.50 ± 1.61 | 8.25 ± 2.25 | – | – | – | – |
| **CoF-CoT** | **57.67 ± 2.75** | **23.47 ± 4.09** | 14.33 ± 1.52 | **9.00 ± 1.00** | **61.50 ± 4.93** | **30.12 ± 3.93** | **15.00 ± 1.32** | **11.00 ± 1.61** |
| | MASSIVE | | | | | | | |
| Direct Prompt | 72.50 ± 4.58 | 33.24 ± 3.34 | 24.17 ± 3.79 | 20.67 ± 3.28 | 75.17 ± 0.58 | 42.36 ± 2.98 | 29.00 ±5.39 | 24.50 ± 4.07 |
| CoT | 71.83 ± 2.57 | 36.32 ± 1.94 | 24.50 ± 2.29 | 21.66 ± 3.40 | 76.83 ± 3.82 | 44.89 ± 2.50 | 31.33 ± 0.87 | 25.83 ± 2.25 |
| SC-CoT | 73.05 ± 1.27 | 37.06 ± 2.54 | 27.16 ±3.21 | 22.50 ± 2.65 | 77.33 ± 2.89 | 47.02 ± 4.60 | 34.00 ± 3.21 | 27.16 ± 3.50 |
| ComplexCoT | 73.66 ± 3.65 | 37.64 ± 3.51 | 25.83 ± 2.25 | 22.16 ± 2.51 | 77.83 ± 1.83 | 46.59 ± 2.43 | 36.50 ± 2.89 | 28.00 ± 3.69 |
| Least-to-Most | 72..83 ± 4.65 | 37.62 ± 1.69 | 31.50 ± 1.53 | 26.50 ± 1.26 | 77..00 ± 3.28 | 45.93 ± 3.99 | 32.50 ± 4.09 | 29.00 ± 5.11 |
| Plan-and-Solve | 69.33 ± 2.47 | 38.07 ± 2.07 | 32.00 ± 1.26 | **29.00 ± 1.26** | – | – | – | – |
| **CoF-CoT** | **89.00 ± 2.29** | **38.66 ± 3.25** | **33.17 ± 4.04** | 25.50 ± 2.64 | **92.00 ±2.29** | **47.06 ± 4.63** | **37.50 ± 1.89** | **29.50 ± 3.12** |

coarse-grained intent label, fine-grained slot labels and slot values. Further details of Logic Form's structure and its connections with multi-grained NLU tasks are provided in the Appendix A.

Our multi-step reasoning is designed in the following sequential order:

1. **Generate AMR**: Given the input utterance, LLMs generate the AMR structured representation (Banarescu et al., 2013). The representation is preserved in the Neo-Davidsonian format as demonstrated in Figure 1,2. Each node in AMR graph refers to a concept, including entity, noun phrase, pre-defined frameset or special keyword. Edges connecting two nodes represent the relation types.

2. **Generate Intent**: In this step, LLMs generate coarse-grained intent label prediction when conditioned on the given input and its corresponding AMR Graph. AMR concepts could provide additional contexts to ambiguous utterances, leading to improved ability to recognize the correct intents.

3. **Generate Slot Values**: In this stage, to generate the fine-grained slot values existent in the input utterance, besides the utterance itself, prompts for LLMs are conditioned on the generated AMR structure and predicted intent label. As AMR graph captures the essential concepts existent in the utterance while abstracting away syntactic idiosyncrasies of the utterance, it can help extract the important concepts mentioned in the utterances. In order to further couple the connections between slots and intents (Zhang et al., 2019; Wu et al., 2020), predicted intents from the Step 2 are also concatenated to construct input prompts for Step 3.

4. **Generate Slot Value, Slot Type pairs**: After obtaining slot values, LLMs label each identified slot value when given the slot vocabulary. Simi-

lar to Step 3, we condition the generated output with the predictions from previous steps, including AMR and intent. Both AMR and intent provide additional contexts for slot type predictions of the given slot values besides the input utterance.

5. **Generate Logic Form**: The last step involves aggregating the predicted intents together with sequences of slot type and slot value pairs to construct the final Logic Form predictions.

## 4 Experiments

### 4.1 Datasets & Preprocessing

We evaluate our proposed framework on two multi-domain NLU datasets, namely MTOP (Li et al., 2021) and MASSIVE (Bastianelli et al., 2020; FitzGerald et al., 2022). As the innate capability of language understanding is best represented via the robustness across different domains, we evaluate the frameworks under low-resource multi-domain settings, including zero-shot and few-shot. Details of both datasets are provided in Appendix B.

To provide a comprehensive evaluation for coarse-grained, fine-grained NLU tasks, as well as the interactions between the two, we conduct an extensive study on both NLU and Semantic Parsing metrics, including: Slot F1-score, Intent Accuracy, Frame Accuracy, Exact Match. Intent Accuracy assesses the performance on coarse-grained sentence-level tasks, while Slot F1 metric evaluates the performance on more fine-grained token-level tasks. The computation of Frame Accuracy and Exact Match captures the ability to establish the accurate connections between sentence-level and token-level elements. For more details of individual metric computation from the Logic Form, we refer readers to (Li et al., 2021).

To conduct the evaluation with efficient API

calls, following (Khattab et al., 2022), we construct test sets by randomly sampling 200 examples covering a set of selected domains, namely test domains. We repeat the process with 3 different seeds to generate 3 corresponding test sets. Reported performance is the average across 3 different seed test sets with standard deviations. For few-shot settings, we randomly select a fixed k samples from a disjoint set of domains, namely train domains. These samples are manually annotated with individual step labels as commonly conducted by other in-context learning CoT approaches (Wei et al., 2022; Wang et al., 2022). Additional implementation details as well as the prompt design and sample outputs are provided in Appendix C and D respectively.

## 4.2 Baseline

**Chain-of-Thought (CoT) Approach Comparison**   We compare our proposed method with the current relevant state-of-the-art CoT approaches:

• **Direct Prompt**: Naive prompting to generate the Logic Form given the intent and slot vocabulary.

• **CoT** (Wei et al., 2022): Automatic generation of series of intermediate reasoning steps from LLMs

• **SC-CoT** (Wang et al., 2022): Enhanced CoT via majority voting among multiple reasoning paths.

• **Complex-CoT** (Fu et al., 2022) : Enhanced CoT by selecting and measuring the consistency of the most complex samples. In our case, we leverage the longest output as the complexity measure.

• **Least-to-Most** (Zhou et al., 2022) : Enhanced CoT by first automatically decomposing the in-hand problems into series of simpler sub-problems, and then solving each sub-problem sequentially.

• **Plan-and-Solve** (Wang et al., 2023) : Enhanced CoT by guiding LLMs to devise the plan before solving the problems by prompting *"Let's first understand the problem and devise a plan to solve the problem. Then, let's carry out the plan and solve the problem step by step."*

**Fine-tuning (FT) Approach Comparison**   As one of the early studies in leveraging LLM for NLU tasks, we also conduct additional comparisons with traditional FT approaches. Specifically, we leverage RoBERTa PLM (Zhuang et al., 2021) with joint Slot Filling and Intent Detection objectives (Li et al., 2021) as the FT model. Unlike LLM, traditional FT operates under closed-world assumption which requires sufficient data to learn domain-specific and domain-agnostic feature extraction in multi-domain settings. For a fair comparison with LLM, we impose an essential constraint that there

Table 2: Comparison between FT and LLM approaches on MTOP dataset.

| Method | Assumption | Intent Acc | Slot F1 | Frame Acc | Exact Match |
|---|---|---|---|---|---|
| RoBERTa FT | Supervised | 67.19 ± 2.90 | 75.17 ± 1.08 | 43.57 ± 4.18 | 36.10 ± 1.08 |
| RoBERTa FT | ZSL | 0 | 12.68 ± 1.25 | 0 | 0 |
| RoBERTa FT | FSL | 0 | 13.75 ± 1.22 | 0 | 0 |
| **CoF-CoT** | **ZSL** | **57.67 ± 2.75** | **23.47 ± 4.09** | **14.33 ± 1.52** | **9.00 ± 1.00** |
| **CoF-CoT** | **FSL** | **61.50 ± 4.93** | **30.12 ± 3.93** | **15.00 ± 1.32** | **11.00 ± 1.61** |

Table 3: Ablation study on the effectiveness of different structured representations on MTOP dataset under zero-shot settings. *CP, DP, AMR* denote *Constituency Parsing, Dependency Parsing and Abstract Meaning Representation* respectively.

| | Intent Acc | Slot F1 | Frame Acc | Exact Match |
|---|---|---|---|---|
| CoT (w/o structure) | 57.16 ± 3.69 | 17.50 ± 2.92 | 12.16 ± 1.61 | 4.67 ± 3.33 |
| CP-CoT | 57.33 ± 3.25 | 19.34 ± 3.34 | 13.16 ± 1.04 | 5.50 ± 1.32 |
| DP-CoT | 57.50 ± 3.01 | 17.83 ± 2.53 | 12.67 ± 1.04 | 5.83 ± 2.08 |
| **AMR-CoT** | **57.67 ± 2.75** | **23.47 ± 4.09** | **14.33 ± 1.52** | **9.00 ± 1.00** |

exist no overlapping domains between train and test domains under ZSL and FSL setting for both FT and CoT approaches. This leads to 3 different scenarios for FT approaches, including:

• **Fully Supervised**: Samples sharing similar domains with test sets are used for training.

• **ZSL**: We utilize samples from domains different from test domains for training.

• **FSL**: We leverage samples from domains different from test domains in conjunction with a fixed number of k-shot test domain samples.

## 5   Result & Discussion

As observed in Table 1, our proposed CoF-CoT achieves state-of-the-art performance across different evaluation metrics on MASSIVE and MTOP datasets under both zero-shot and few-shot settings. The performance gain over the most competitive baseline is more significant in terms of Intent Accuracy (25% and 15.34% improvements on MTOP and MASSIVE respectively in zero-shot settings). Additional case studies presented in Appendix E further demonstrate the effectiveness of CoF-CoT.

In addition, we observe consistent improvements of different CoT variants over the Direct Prompt. It implies that CoT prompting allows the model to reason over multiple steps and learn the connections between different NLU tasks more effectively.

In comparison with MASSIVE, performance of all methods is significantly lower on MTOP. It is mainly due to the more complex Logic Form structures existent in MTOP. It is noticeable that MASSIVE datasets contain samples of fewer average number of slots, leading to significantly better performance on Semantic Parsing tasks (i.e. Frame Accuracy and Exact Match).

Our CoF-CoT shares certain degrees of similarities with Least-to-Most (Zhou et al., 2022), Plan-and-Solve prompting (Wang et al., 2023). How-

Table 4: Ablation study of step ordering on MASSIVE dataset. CoF and FoC denote Coarse-to-Fine-grained and Fine-to-Coarse-grained order respectively.

| Method | Assumption | Intent Acc | Slot F1 | Frame Acc | Exact Match |
|---|---|---|---|---|---|
| Random-CoT | Random Order | 80.67 ±3.60 | 27.14 ± 2.47 | 26.50 ±1.80 | 16.50 ± 1.04 |
| FoC-CoT | FoC order | 83.00 ± 2.88 | 32.11 ± 2.50 | 28.50 ± 3.21 | 18.00 ± 3.50 |
| CoF-CoT (w/o step 1) | No AMR | 81.50 ± 4.36 | 33.68 ± 2.40 | 27.50 ± 2.65 | 18.00 ± 0.76 |
| CoF-CoT (w/o step 2) | No intent | 78.17 ± 4.80 | 27.66 ± 1.93 | 23.50 ± 2.78 | 14.50 ± 2.25 |
| CoF-CoT (w/o step 3) | No separate KP | 82.33 ± 1.04 | 34.63 ± 3.10 | 32.83 ± 2.47 | 23.00 ± 1.80 |
| CoF-CoT (w/o step 4) | No separate slot prediction for KP | 79.17 ± 4.01 | 32.92 ± 5.02 | 31.50 ± 3.50 | 21.83 ± 3.17 |
| CoF-CoT (w/o step 3+4) | No separate slot prediction | 81.33 ± 4.19 | 31.31 ±3.77 | 27.67 ± 5.34 | 21.00 ± 4.92 |
| **CoF-CoT** | **CoF order (Full)** | **89.00 ± 2.29** | **38.66 ± 3.25** | **33.17 ± 4.04** | **25.50 ± 2.64** |
| - Conditioning | No domain | 84.50 ± 2.75 | 36.80 ± 2.08 | 32.50 ± 1.73 | 24.83 ± 0.58 |

ever, unlike the two aforementioned baselines that rely heavily on the existent pre-trained knowledge of LLMs, CoF-CoT provides a controllable number of sequential steps and conditioning inputs for each step, allowing for flexible adaptations and customizations to future downstream tasks that LLMs might not be familiar with.

**Comparison with FT**    Under ZSL and FSL settings, the FT model suffers from the aforementioned domain gap issues. Specifically, as observed in Table 2, since there exist minimal overlapping intent labels between train and test domains, without sufficient data in ZSL and FSL settings, the FT approaches are unable to learn transferable multi-domain features, leading to 0 performance in Intent Accuracy. This behavior also results in 0 performance for both Frame Acc and Exact Match as the correct intents are the prerequisites for correct semantic frame and exact match metrics. On the other hand, *Fully supervised FT* approach acquires domain-specific knowledge of target domains from training data and performs the best across different evaluation metrics. However, this assumption does not directly match ZSL/FSL settings in which LLMs are currently evaluated.

**Impact of Structured Representation**   Besides AMR Graph, there exist other structured representations that directly link to semantic and syntactic understanding of utterances, including DP,CP. Our empirical study presented in Table 3 reveals that AMR-CoT unanimously achieves the best performance, demonstrating its effectiveness in capturing the diversity of input utterances when compared with other structured representations.

**Impact of Step Order**   To further understand the importance of the designed CoF step order, we conduct additional ablation studies on 3 different scenarios: (1) random ordering (step 3→1→2→4→5), (2) Fine-to-Coarse (FoC) ordering (step 1→3→4→2→5), and (3) CoF ordering

with hypothetical individual step removal. Table 4 demonstrates CoF logical ordering yields the best performance with significant improvements on the challenging Exact Match metrics (9.00 and 7.50 points of improvement over random and FoC respectively). Random, FoC ordering together with CoF ordering with missing individual steps neglect the natural connections of problem-solving from high-level (coarse-grained) to low-level (fine-grained) tasks, leading to worse performance across different metrics. For CoF-CoT, when step 1 or step 2 is removed (no AMR or intent information), we observe the most significant performance decrease, implying the essence of coarse-grained knowledge for LLMs to solve the later sequential steps.

**Impact of Conditioning**   The major advantage of our multi-step reasoning is the ability to explicitly condition the prior predictions in later steps. As observed in Table 4, conditioning prior knowledge in multi-step reasoning improves the overall performance of CoF-CoT across different metrics with the most significant gain in Intent Accuracy (+4.50%). This observation implies the importance of conditioning the appropriate information on CoT for an improved performance of LLMs under challenging zero-shot multi-domain settings.

## 6   Conclusion

In this work, we conduct a preliminary study of LLMs' capability in multi-grained NLU tasks of Dialogue systems. Moreover, motivated by CoT, we propose a novel CoF-CoT approach aiming to break down NLU tasks into multiple reasoning steps where (1) LLMs can learn to acquire and leverage concepts from different granularities of NLU tasks, (2) additional AMR structured representation can be integrated and leveraged throughout the multi-step reasoning. We empirically demonstrate the effectiveness of CoF-CoT in improving LLMs capability in multi-grained NLU tasks under both zero-shot and few-shot multi-domain settings.

## Limitations

Our empirical study is restricted to English NLU data. It is partially due to the existent English-bias of Abstract Meaning Representation (AMR) structure (Banarescu et al., 2013). We leave the adaptation of the CoF-CoT to multilingual settings (Nguyen and Rohrbaugh, 2019; Qin et al., 2022; Nguyen et al., 2023b) as future directions for our work.

Our work is empirically studied on the Flat Logic Form representation. In other words, Logic Form only includes one intent followed by a set of slot sequences. There are two major rationales for our empirical scope. Firstly, as the early preliminary study on multi-grained NLU tasks which unify both Semantic Parsing and NLU perspectives, we design a small and controllable scope for the experiments. Secondly, as most NLU datasets including MASSIVE (FitzGerald et al., 2022) are restricted to single-intent utterances, Flat Logic Form is a viable candidate reconciliating between traditional NLU and Semantic Parsing evaluations. We leave explorations on the more challenging Nested Logic Form where utterances might contain multiple intents for future work.

## 7 Acknowledgement

We thank the anonymous reviewers for their constructive feedback which we incorporated in the final version of this manuscript.

This work is supported in part by NSF under grant III-2106758.

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

Table 5: Details of MTOP and MASSIVE datasets

| Dataset | MTOP | MASSIVE |
|---|---|---|
| # Domains | 11 | 18 |
| # Train Domains | 8 | 14 |
| # Test Domains | 3 | 4 |
| # Intents | 117 | 60 |
| # Slots | 78 | 55 |
| Sentence Length | $6.14 \pm 2.30$ | $6.34 \pm 2.94$ |
| # Slots per sample | $1.87 \pm 0.81$ | $0.73 \pm 0.63$ |

# A  Connections between Semantic Parsing and NLU Tasks via Logic Form

Logic Form not only captures the coarse-grained intent labels and fine-grained slot labels of the utterances but also encapsulates the implicit connections between slots and intents.

As observed in Table 7, Logic Form is constructed as the flattened representation of the dependency structure between intents and slot sequences. Semantic Frame constructed as intent type(s) followed by a sequence of slot types can be directly extracted from the Logic Form. In addition, via the Logic Form, the coarse-grained intent label CREATE_REMINDER, fine-grained slot TODO, DATE_TIME labels together the respective slot values (*message Mike*, *at 7pm tonight*) can all be extracted and converted to appropriate format (i.e. BIO format as the traditional sequence labeling ground truths (Zhang et al., 2019)). Therefore, Logic Form can be considered the unified label format to bridge the gap between Semantic Parsing (Li et al., 2021; Xie et al., 2022) and traditional Intent Detection and Slot Filling tasks in NLU systems (Xia et al., 2020; Nguyen et al., 2020; Casanueva et al., 2022).

# B  Dataset Details

We provide the details of MTOP and MASSIVE datasets in Table 5. As compared to MASSIVE, MTOP dataset not only contains more slot types and intent types but also tends to cover more slot types per sample in the Logic Form. This challenging characteristic explains the consistent lower performance across all methods on MTOP when compared to MASSIVE as observed in Section 5.

# C  Implementation Details

As the proposed step-by-step reasoning can be applied to any LLMs, our proposed method is LLM-agnostic which is empirically studied in Appendix F. For simplicity and consistency, in our main empirical study, we leverage *gpt-3.5-turbo* from Ope-

nAI as the base LLM model. Following (Wang et al., 2022), we set the decoding temperature T=0.7 and number of outputs n=10.

As domain names provide essential clues for language models in multi-domain settings for multi-grained NLU tasks (Zhou et al., 2023), to safeguard the fairness in baseline comparisons, we consistently include the *domain name* in the input prompts for all baselines unless stated otherwise. Specifically, the only exception is presented in Table 4 for *CoF-CoT(CoF order)-Conditioning*.

For few-shot (i.e. k-shot) learning settings, we randomly sample k examples and manually prepare the necessary labels for different baseline variants. We experiment with k=5 in our empirical study.

**Domains of Demonstration Samples** To replicate a more realistic scenario where the domains of k-shot demonstration samples are generally unknown, we assume that k-shot demonstration samples come from different domains from the test samples. The relaxation of constraints on the assumption regarding the domain similarity between demonstration samples and test samples allows for broader applications and encourages LLMs to accumulate and extract the true semantic knowledge from k-shot demonstrations and avoid overfitting any specific domains. For completeness, we also conduct additional empirical studies to compare the FSL performance of CoF-CoT under both scenarios: (1) k demonstration samples are from the same domain as test samples, (2) k demonstrations are drawn from different domains from the test samples. As observed in Table 6, additional constraint of similar domains between k-shot demonstration samples and test samples leads to improvements in the evaluation performance across NLU and Semantic Parsing tasks. This might be intuitive since LLMs can extract domain-relevant information from the given k domain-similar samples to assist with inference process on test samples.

## D Prompt Design

Prompts for individual steps of our CoF-CoT are presented in Figure 3. Additional output samples are also provided in Figure 4.

## E Qualitative Case Study

We present additional Qualitative Case Study comparing the outputs between different baseline methods and our proposed CoF-CoT in Figure 5.

As observed in Figure 5, our CoF-CoT provides the predictions closest to the ground truth while other baselines struggle to (1) generate the correct intent type (i.e. GET_DATE_TIME_EVENT intent type from Direct Prompt in comparison with GET_EVENT intent from ground truth) (2) identify the correct slot values (i.e. *everything* slot value generated from CoT), (3) generate the correct slot type for the corresponding slot values. (i.e. EVENT_TYPE slot type for *music festivals* slot values from Complex-CoT instead of CATEGORY_EVENT slot type).

## F LLM-Agnostic Capability

Our proposed CoF-CoT is LLM-agnostic since the focus of the work is on the prompt design, which can be applied to any LLMs. As most LLMs rely on the high quality of the designed prompts, our proposed CoF-CoT prompt design can be used as input to any LLMs for zero-shot and in-context learning settings. This is also similarly observed in CoT (Wei et al., 2022), SC-CoT (Wang et al., 2022) and other comparable CoT methods. For further clarification, we report additional empirical results of our proposed CoF-CoT applied to both of the backbone PaLM (Chowdhery et al., 2022) and GPT3.5 LLMs on the MTOP dataset under both ZSL and FSL settings in Table 8. As observed in Table 8, CoF-CoT prompting consistently outperforms the two backbone LLMs across all NLU and Semantic Parsing tasks, demonstrating both the effectiveness and LLM-agnostic capability of our proposed CoF-CoT.

Table 6: FSL Results of CoF-CoT with k-shot demonstration samples selected from different and similar domains in comparison with domains of test samples on MTOP dataset.

| Method | Assumption | Intent Acc | Slot F1 | Frame Acc | Exact Match |
|---|---|---|---|---|---|
| CoF-CoT | k domain-different samples | $61.50 \pm 4.93$ | $30.12 \pm 3.93$ | $15.00 \pm 1.32$ | $11.0 \pm 1.61$ |
| CoF-CoT | k domain-similar samples | $70.00 \pm 1.33$ | $38.16 \pm 5.42$ | $20.50 \pm 2.00$ | $15.00 \pm 1.00$ |

Table 7: Sample utterance with its Logic Form under both Semantic Parsing and NLU tasks' metrics. // denotes the separation between tokens of the given utterance.

| | Metric | Granularity Level | Format | Ground Truth |
|---|---|---|---|---|
| Input Sentence | – | – | – | Set // up// a // reminder // to // message // mike // at // 7pm // tonight |
| Logic Form | – | – | – | [IN:CREATE_REMINDER [SL:TODO: message mike] [SL:DATE_TIME: at 7pm tonight]] |
| NLU Tasks | Intent Accuracy | Coarse-grained | Intent Label | IN:CREATE_REMINDER |
| | Slot F1 | Fine-grained | BIO Slot Sequence | O // O // O // O // O // B-TODO // I-TODO // B-DATE_TIME // I-DATE_TIME // I-DATE_TIME |
| Semantic Parsing Tasks | Frame Accuracy | Both | Logic Form | IN:CREATE_REMINDER-SL:TODO-SL:DATE_TIME |
| | Exact Match | Both | Logic Form | [IN:CREATE_REMINDER [SL:TODO: message mike] [SL:DATE_TIME: at 7pm tonight]] |

**Step 1:** Given the utterance and its domain, generate a single corresponding Abstract Meaning Representation (AMR) Graph in the Neo-Davidsonian format. The format involves :ARG and :op relations.

Utterance: {utterance}
Domain: {domain}

AMR Graph: {AMR}
-----------------------------------------------------------------------------------------------------------------------------------------------
**Step 2:** Given the utterance and its domain, and its AMR Graph, select one of the following in the Intent Vocabulary as the intent type for the utterance.

Utterance: {utterance}
Domain: {domain}
AMR Graph: {AMR}]
Intent Vocabulary: {intent_vocab}

Intent:{intent}
-----------------------------------------------------------------------------------------------------------------------------------------------
**Step 3:** Based on the utterance, domain, its AMR Graph and its intent, generate key phrases for the utterance. Key phrases can be made up from multiple AMR concepts.
Each word in key phrases must exist in the given utterance. Each word in the utterance appears in only one key phrase. Key phrases need to contain consecutive words in the given utterance.
Key phrases do not need to cover all words in the utterance. Return a list of key phrases separated by commas.

Utterance: {utterance}
Domain: {domain}
AMR Graph: {AMR}
Intent: {intent}

Key phrases: {key_phrase}
-----------------------------------------------------------------------------------------------------------------------------------------------
**Step 4:** Given the slot vocabulary, utterance, its domain, its AMR Graph and its intent, identify the corresponding slot type as one of the types in the slot vocabulary for each key phrase.
Return the list of key phrases and their corresponding slot types in the following format: (key_phrase, slot_type) separated by commas.
If none of the slot types in the vocabulary fits, return the slot type as O.

Slot Vocabulary: {slot_vocab}
Utterance: {utterance}
Domain: {domain}
AMR Graph: {AMR}
Intent: {intent}

(Key phrase, Slot Type) pairs: {slot_pair}
-----------------------------------------------------------------------------------------------------------------------------------------------
**Step 5:** Given the utterance, domain, its intent type, its slot type and slot value pairs in (slot_type, slot_value) format,
generate logic form of the utterance in the format of [IN:___ [SL:___] [SL:___]] where IN: is followed by an intent type and SL: is followed by a slot type and slot value pair separated by white space.
The number of [SL: ] is unlimited. The number of [IN: ] is limited to 1.

Utterance: {utterance}
Domain: {domain}
Intent: {intent}
(Key phrase, Slot Type) pairs: {slot_pair}

Logic Form: {lf}

Figure 3: CoF-CoT Prompt Design Template. {.} denotes the placeholder argument.

Table 8: Experimental results on MTOP dataset under zero-shot few-shot multi-domain settings with different LLM backbone architectures (PaLM (Chowdhery et al., 2022) and GPT3.5).

| | MTOP | | | | | | | |
|---|---|---|---|---|---|---|---|---|
| Model | Zero-shot | | | | Few-shot | | | |
| | NLU | | Semantic Parsing | | NLU | | Semantic Parsing | |
| | Intent Acc | Slot F1 | Frame Acc | Exact Match | Intent Acc | Slot F1 | Frame Acc | Exact Match |
| PaLM | $16.67 \pm 2.52$ | $7.24 \pm 1.00$ | $3.17 \pm 0.76$ | $1.17 \pm 0.76$ | $48.83 \pm 4.54$ | $14.24 \pm 1.58$ | $4.17 \pm 2.02$ | $2.33 \pm 0.76$ |
| **PaLM + CoF-CoT** | $\mathbf{42.33 \pm 3.33}$ | $\mathbf{13.73 \pm 2.88}$ | $\mathbf{4.01 \pm 0.15}$ | $\mathbf{3.50 \pm 1.32}$ | $\mathbf{57.17 \pm 3.79}$ | $\mathbf{21.47 \pm 3.79}$ | $\mathbf{10.33 \pm 3.06}$ | $\mathbf{6.67 \pm 2.75}$ |
| GPT3.5 | $31.50 \pm 1.80$ | $21.84 \pm 2.83$ | $8.33 \pm 1.44$ | $6.00 \pm 1.32$ | $51.33 \pm 3.40$ | $28.35 \pm 3.24$ | $11.00 \pm 1.80$ | $8.33 \pm 1.00$ |
| **GPT3.5 + CoF-CoT** | $\mathbf{57.67 \pm 2.75}$ | $\mathbf{23.47 \pm 4.09}$ | $\mathbf{14.33 \pm 1.52}$ | $\mathbf{9.00 \pm 1.00}$ | $\mathbf{61.50 \pm 4.93}$ | $\mathbf{30.12 \pm 3.93}$ | $\mathbf{15.00 \pm 1.32}$ | $\mathbf{11.00 \pm 1.61}$ |

**Sample 1:**
**{utterance}** = show me dates for music festivals in 2018
**{ground_truth}** = [IN:GET_EVENT [SL:CATEGORY_EVENT music festivals ] [SL:DATE_TIME in 2018 ]]
**{domain}** = event

**{AMR}** = (c/ show-01
  :ARG1 (m /me)
  :ARG2 (d/ date-entity
    :mod (f/ music-festival)
    :year 2018))
**{intent}** = GET_EVENT
**{key phrase}** = Show me, dates, music fesitval, in 2018
**{Key phrase, Slot Type}** = (Show me: O), (dates: O), (music festival: CATEGORY_EVENT), (in 2018: DATE_TIME)
**{lf}** = [IN:GET_EVENT [SL:CATEGORY_EVENT music festivals ] [SL:DATE_TIME in 2018 ]]
--------------------------------------------------------------------------------------------------------------------------------------------------
**Sample 2:**
**{utterance}** = Set my timer for my tabata workout.
**{ground_truth}** = [IN:CREATE_TIMER [SL:METHOD_TIMER timer ] [SL:TIMER_NAME tabata workout ] ]
**{domain}** = timer

**{AMR}** = (set-01
  :ARG0 (I)
  :ARG1 (timer-02
    :ARG0 (my-03)
     :op1 (workout-05
      :ARG0 (my-04)
      :ARG1 (tabata-06))))
**{intent}** = CREATE_TIMER
**{key phrase}** = Set my timer, tabata workout
**{Key phrase, Slot Type}** = (Set my timer: METHOD_TIMER), (tabata workout: TIMER_NAME)
**{lf}** = [IN:CREATE_TIMER [SL:METHOD_TIMER Set my timer ] [SL:TIMER_NAME tabata workout ]]

Figure 4: Sample output from our CoF-CoT. {.} denotes the placeholder corresponding to template in Figure 3.

Utterance 1: Show me dates for music festivals in 2018
**Ground Truth:** **[IN:GET_EVENT [SL:CATEGORY_EVENT music festivals ] [SL:DATE_TIME in 2018 ]]**
--------------------------------------------------------------------------------------------------------------------------------------------------
Direct Prompt: [IN:GET_DATE_TIME_EVENT [SL:TITLE_EVENT music festivals ] [SL:DATE_TIME:2018 ]]
CoT: [IN:GET_EVENT [SL:DATE_TIME 2018 ] [SL:TITLE_EVENT music festivals ]]
SC-CoT: [IN:GET_EVENT [SL:DATE_TIME 2018 ] [SL:TITLE_EVENT music festivals ]]
Complex-CoT: IN:GET_DATE_TIME_EVENT [SL:EVENT_TYPE music festivals ] [SL:EVENT_YEAR 2018 ]]
--------------------------------------------------------------------------------------------------------------------------------------------------
**CoF-CoT:** **[IN:GET_EVENT [SL:CATEGORY_EVENT music festivals ] [SL:DATE_TIME in 2018 ]]**

Utterance 2: I want to know everything that breaks about the dam breaking in california
**Ground Truth:** **[IN:NEWS_QUERY [SL:PLACE_NAME california ]]**
--------------------------------------------------------------------------------------------------------------------------------------------------
Direct Prompt: [IN: WEATHER_QUERY] [SL: PLACE_NAME california ] [SL: EVENT_NAME dam breaking ]]
CoT: IN: NEWS_QUERY [SL: WEATHER_DESCRIPTOR everything ] [SL: PLACE_NAME california ]]
SC-CoT: [IN:NEWS_QUERY [SL:NEWS_TOPIC dam breaking in california ]]
Complex-CoT: [IN:NEWS_QUERY [SL:NEWS_TOPIC dam breaking in california ]]
--------------------------------------------------------------------------------------------------------------------------------------------------
**CoF-CoT:** **[IN: NEWS_QUERY [SL: PLACE_NAME california ]]**

Figure 5: Qualitative Case Study among baseline variants and the proposed CoF-CoT. Ground Truth is shown in red.