# OpenReview forum: "CoF-CoT: Enhancing Large Language Models with Coarse-to-Fine Chain-of-Thought Prompting for Multi-domain NLU Tasks"
_EMNLP/2023/Conference — EMNLP 2023 Main_

### Official Review · Reviewer_ZdgB · 2023-07-28

**Soundness:** 4

**Excitement:**

3: Ambivalent: It has merits (e.g., it reports state-of-the-art results, the idea is nice), but there are key weaknesses (e.g., it describes incremental work), and it can significantly benefit from another round of revision. However, I won't object to accepting it if my co-reviewers champion it.

**Paper Topic And Main Contributions:**

This paper propose a novel approach called CoF-CoT, which is a modification of CoT that enables multi-step reasoning for natural language understanding in dialogue. The core idea of CoF-CoT is to prompt the large language model (LLM) to perform multiple reasoning steps from coarse to fine-grained levels. This includes generating abstract meaning representation, generate intention, generate slot values, generate slot value-type pairs, and generate logic form. The paper evaluates CoF-CoT on two datasets and shows that it outperforms previous CoT variants. The paper also analyzes the impact of different structured representations and step orders on the reasoning performance and demonstrates the benefits of a coarse-to-fine-grained reasoning process for LLMs.

**Questions For The Authors:**

1. As mentioned in weakness, will the proposed CoF-CoT still work when the intermediate ground truths are not available?
2. What is the importance of each reasoning step (e.g., What if I remove one of the first four step)?
3. Have you conducted any experiments for the selection of k-shot examples? For example, what if I select examples that are within the same domain of the test example?

**Reasons To Accept:**

1. The idea of using CoT for multi-step reasoning on LLMs is a novel an interesting problem, and the paper propose a success exploration in the NTU domain.
2. According to the paper, CoF-CoT has a significant improvement in performance compared to other variants of CoT.

**Reasons To Reject:**

For MTOP and MASSIVE, ground truths of intermediate reasoning results are provided, which make the few-shot learning of CoF-CoT possible. However, as shown in the paper, currently all the CoT methods (including CoF-CoT) perform poorly under zero-shot setting, the method may have limited ability when a new scenario where the intermediate ground truths are not available.

**Reproducibility:**

4: Could mostly reproduce the results, but there may be some variation because of sample variance or minor variations in their interpretation of the protocol or method.

**Reviewer Confidence:**

3: Pretty sure, but there's a chance I missed something. Although I have a good feel for this area in general, I did not carefully check the paper's details, e.g., the math, experimental design, or novelty.

---

> ### Author Rebuttal · Authors · 2023-08-29
>
> Thank you for your comments. We would like to clarify a few major points:
>
> **(1) Intermediate Ground Truth requirements (Q1, RR):** Generally speaking, our CoF-CoT does not require intermediate ground truths to work.
>
> * For Zero-shot Learning (ZSL), the input is the utterance sample. Each step (from step 2) of our proposed CoF-CoT leverages the output from the previous step as an additional source of input information. This is simply the benefits when we leverage multi-step reasoning. Throughout this process, no intermediate ground truth labels (i.e. Abstract Meaning Representation (AMR) Graph) are needed or provided.
>
> * For Few-shot Learning (FSL),  the input is the utterance together with a fixed number of k-shot demonstration samples  (k=5 in our study) for in-context learning. Similar to other CoT approaches, we prepare labels of individual steps for a fixed number of k-shot demonstration samples in order to represent the in-context samples in a multi-step reasoning process. In our case, we only need to generate AMR Graph for k demonstration samples as presented in Line 513-516. This procedure is simple and efficient in our case since the k-shot demonstration samples are fixed and the AMR graph can be auto-generated by external tools.
>
>
> The gap between ZSL and FSL evaluation does not purely come from the inclusion of intermediate ground truths. It is also due to the introduction of in-context demonstration samples, which helps (1) LLMs familiarize themselves with the downstream tasks in terms of expected output, format, etc. (2) LLMs acquire multi-step reasoning procedure knowledge from in-context samples and apply it to test samples.
>
> **(2) Importance of individual reasoning steps (Q2):** We conducted ablation studies to study the importance of multiple steps in our framework. Firstly, removing all of the first four steps will be equivalent to Direct Prompting as shown in Table 1.  Removing step 1 is equivalent to discarding the leverage of AMR Graph, which is demonstrated in Table 3. As the remaining steps are more coherent and produce necessary outputs for the aggregation step, instead of removing these steps, we verified the importance of the designed sequential ordering  by conducting additional experiments in (1) reversing the CoF to FoC order, and (2) random ordering as presented in Table 3 and Line 263-275.
>
>  **(3) K shot demonstration/ example selection (Q3):** Although we hypothesize and empirically validate in Table A that additional constraint of similar domains between k-shot demonstration samples and test samples could help further improve the evaluation performance, this setting might be unrealistic as the domains of test samples are generally unknown. For broader applications, we relax this constraint in our settings so that LLMs need to accumulate and extract the true semantic knowledge from k-shot demonstrations and avoid overfitting any specific domains. In our setting, k-shot demonstrations are from domains that are different from the test samples.
>
> |  Method | Assumption | Intent Acc | Slot F1 |Frame Acc | Exact Match |
> |-----------|-----------|-----------|-----------|-----------|-----------
> | CoF-CoT | k domain-different samples| 61.50 $\pm$ 4.93 | 30.12 $\pm$ 3.93 |  15.00 $\pm$ 1.32| 11.00 $\pm$ 1.61|
> | CoF-CoT | k domain-similar samples| 70.00$\pm$ 1.33 | 38.16 $\pm$ 5.42 |  20.50 $\pm$ 2.00 | 15.00 $\pm$ 1.00|
>
> Table A: FSL results of CoF-CoT with k-shot demonstration samples selected from different and similar domains in comparison with test samples on MTOP dataset.
>
> **Reference:** RR1: Reason to Reject 1, Q1: Question 1

---

### Official Review · Reviewer_ajre · 2023-08-04

**Typos Grammar Style And Presentation Improvements:** 1) Line 34
**Soundness:** 4

**Excitement:**

3: Ambivalent: It has merits (e.g., it reports state-of-the-art results, the idea is nice), but there are key weaknesses (e.g., it describes incremental work), and it can significantly benefit from another round of revision. However, I won't object to accepting it if my co-reviewers champion it.

**Missing References:**

NA

**Paper Topic And Main Contributions:**

The paper discusses Natural Language Understanding (NLU) in Dialogue systems and the challenges associated with different granularities of NLU tasks. It proposes an adaptation of the Chain-of-Thought (CoT) approach to address multi-grained NLU tasks in a coarse-to-fine-grained sequential reasoning order. The authors also suggest integrating structured knowledge represented via Abstract Meaning Representation (AMR) Graph to enhance the reasoning capabilities of Large Language Models (LLMs) in Dialogue systems.

**Questions For The Authors:**

1) Could the proposed method be combined with self-consistency?

I will update the final score according to the authors responses.

**Reasons To Accept:**

1) The proposed idea is interesting and reasonable.
2) The paper is easy to follow

**Reasons To Reject:**

1) The paper lacks content on related work regarding NLU, Multi-grained NLU, Multi-grained NLU of dialogue systems, and Multi-domain NLU.

2) The related work on CoT is not well written.

3) This paper lacks baselines with full parameter tuning or non-CoT baselines.

4) The paper does not compare to multi-step prompting methods like the few-shot based least-to-most prompting [1] and the zero-shot based plan-and-solve prompting [2].

5) It is not clear how to implement the proposed method in zero-shot and few-shot scenarios.

**Reproducibility:**

4: Could mostly reproduce the results, but there may be some variation because of sample variance or minor variations in their interpretation of the protocol or method.

**Reviewer Confidence:**

2: Willing to defend my evaluation, but it is fairly likely that I missed some details, didn't understand some central points, or can't be sure about the novelty of the work.

---

> ### Author Rebuttal · Authors · 2023-08-29
>
> Thank you for your comments. We address your questions as follows.
>
> **(1) Consistency integration (Q1)**: Our CoF-CoT is independent from the proposal of self-consistency. In other words, it can be combined with self-consistency (by allowing multiple outputs from LLMs and selecting the most consistent one as the final output) or can stand alone by itself. However, as self-consistency is not our main claim or contribution, we focus on verifying the effectiveness of our proposed (1) multi-step reasoning from coarse-grained to fine-grained , (2) integrating structured knowledge from AMR graphs in multi-step reasoning for multi-grained NLU tasks.
>
>
> **(2) Zero-shot and Few-shot Settings setups (RR5):** For zero-shot learning (ZSL), the setup is as mentioned in the test set construction (Line 207-213). Regarding the few-shot learning (FSL) setup, our k-shot setup is presented in Line 513-516. An important note is that for FSL, the domains of k-shot are distinct from the domains of the test sets. We will clarify with additional details in our final manuscript regarding the setups for further transparency.
>
> **(3) Further CoT baselines (RR4):** The major difference between our method and Least-to-Most prompting [1], Plan-and-Solve prompting [2] is that we present a controllable number of sequential steps and controllable conditioning inputs for each sequential step. Both Least-to-Most and Plan-and-Solve prompting rely heavily on the pre-trained knowledge of LLMs, leading to potential hallucinations and worse performance if LLMs are not familiar with the downstream tasks (in our case, multi-grained NLUs). Table A below presents our empirical study on MTOP dataset. As Plan-and-Solve was proposed for ZSL, we only report ZSL performance for this method. We will also update the completed results for the MASSIVE dataset in the final version.
>
> As observed in our empirical study in Table A, under the same setup, our proposed CoF-CoT achieves consistent improvements on NLU tasks and comparable performance with the other prompting techniques on other semantic parsing metrics.
>
> |  | Intent Acc | Slot F1 |Frame Acc | Exact Match | Intent Acc | Slot F1 |Frame Acc | Exact Match |
> |-----------|-----------|-----------|-----------|-----------|-----------|-----------|-----------|----------|
> | Least-to-Most  | 45.67 $\pm$ 0.58 | 21.84 $\pm$ 1.91 | **14.50 $\pm$ 0.50** | 8.00 $\pm$ 0.50 | 49.83 $\pm$ 4.54 | 27.28 $\pm$ 2.41  | **16.00 $\pm$ 0.50**  |8.83 $\pm$ 0.76 |
> | Plan-and-Solve  | 45.00 $\pm$ 4.00 | 22.45 $\pm$ 2.28 |9.50 $\pm$ 1.61 | 8.25 $\pm$ 2.25 |--|--|--|--|
> | **CoF-CoT** |  **57.67 $\pm$ 2.75** | **23.47 $\pm$ 4.09** |  14.33 $\pm$ 1.52| **9.00 $\pm$ 1.00** | **61.50 $\pm$ 4.93** | **30.12 $\pm$ 3.93** |  15.00 $\pm$ 1.32| **11.00 $\pm$ 1.61**|
>
> Table A: Performance of additional relevant CoT baselines and the proposed CoF-CoT on MTOP dataset.
>
>
> **(4) Comparison with fine-tuning (FT) approaches of Pre-trained Language Models (PLM) (RR3):**
> We argue traditional FT (i.e. full parameter training as proposed in [3],[4]) and LLM ZSL/FSL approaches are not directly comparable. Traditional FT operates under closed-world assumption which requires sufficient data to learn domain-specific and domain-agnostic feature extraction in multi-domain settings. Without sufficient data in  ZSL/FSL settings, FT approaches might not be able to capture these features, leading to inferior performance, especially when the domain gap between training and testing is significant. On the other hand, LLM provides an open-domain approach where fine-tuning on specific domains is not needed.
>
>
> We conducted a preliminary study and observed the aforementioned gap between the two approaches. For fair comparison between FT and LLM approaches, our study imposes the constraint that  there exist no overlapping domains between train and test domains under ZSL and FSL setting. We experimented with 3 different scenarios for FT approaches, including:
>
> * *Fully Supervised:* Samples sharing similar domains with test sets are used for training
> * *ZSL:* We use samples from domains different from test domains for training
> * *FSL:* We use samples from domains different from test domains in conjunction with a fixed number of  k-shot test domain samples.
>
> We leverage Roberta PLM [5] with joint Slot Filling and Intent Detection objectives as presented in [3],[4] as FT model. Under ZSL and FSL settings, the FT model suffers from the aforementioned domain gap issues. Specifically, as observed in Table B, since there exist minimal overlapping intent labels between train and test domains, without sufficient data in ZSL and FSL settings, the FT approaches are unable to learn transferable multi-domain features, leading to 0 performance in Intent Accuracy. This behavior also results in 0 performance for both Frame Acc and Exact Match as the correct intents are the prerequisites for correct semantic frame and exact match metrics. On the other hand, Fully supervised FT approach acquires domain-specific knowledge of target domains from training data and performs the best across different evaluation metrics. However, this assumption does not match ZSL/FSL settings.
>
> |  Method | Assumption | Intent Acc | Slot F1 |Frame Acc | Exact Match |
> |-----------|-----------|-----------|-----------|-----------|-----------
> | Roberta FT  | Fully Supervised | 67.19 $\pm$ 2.90 | 75.17 $\pm$ 1.08 |43.57 $\pm$ 4.18 | 36.10 $\pm$ 1.08 |
> | Roberta FT | ZSL| 0| 12.68 $\pm$ 1.25 | 0 | 0|
> | Roberta FT | FSL| 0| 13.75 $\pm$ 1.22 | 0 | 0|
> | GPT3.5 + CoF-CoT | ZSL| 57.67 $\pm$ 2.75 | 23.47 $\pm$ 4.09 |  14.33 $\pm$ 1.52| 9.00 $\pm$ 1.00|
> | GPT3.5 + CoF-CoT | FSL| 61.50 $\pm$ 4.93 | 30.12 $\pm$ 3.93 |  15.00 $\pm$ 1.32| 11.00 $\pm$ 1.61|
>
> Table B: Complete comparison between FT and LLM approaches on MTOP dataset.
>
> **(5) Backgrounds of NLU Tasks and CoT (RR1, RR2):** We will update further details of the NLU and CoT backgrounds in the Related work in our final version.
>
> **Reference:** RR1: Reason to Reject 1, Q1: Question 1
>
> [1] Zhou et al., Least-to-Most Prompting Enables Complex Reasoning in Large Language Models. ICLR 2023.
>
> [2] Wang et al., Plan-and-Solve Prompting: Improving Zero-Shot Chain-of-Thought Reasoning by Large Language Models. ACL 2023.
>
> [3] Li et. al, MTOP: A Comprehensive Multilingual Task-Oriented Semantic Parsing Benchmark. EACL 2021.
>
> [4] FitzGerald et al., MASSIVE: A 1M-Example Multilingual Natural Language Understanding Dataset with 51 Typologically-Diverse Languages. 2022. arXiv preprint arXiv:2204.08582.
>
> [5] Liu et. al, RoBERTa: A Robustly Optimized BERT Pretraining Approach. 2019. arXiv:1907.11692.

---

### Official Review · Reviewer_GkrJ · 2023-08-05

**Typos Grammar Style And Presentation Improvements:** 1. Page 2, line 105 and 117. You only…
**Soundness:** 4

**Excitement:**

3: Ambivalent: It has merits (e.g., it reports state-of-the-art results, the idea is nice), but there are key weaknesses (e.g., it describes incremental work), and it can significantly benefit from another round of revision. However, I won't object to accepting it if my co-reviewers champion it.

**Paper Topic And Main Contributions:**

This paper studies the application of Chain-of-Thought (CoT) to Large Language Models (LLMs) in Natural Language Understanding (NLU). Specifically, they propose Coarse-to-Fine CoT (CoF-CoT) approach to enable LLMs to acquire and leverage essential concepts to solve tasks from different granularities. Moreover, they integrate structured knowledge based on semantics via Abstract Meaning Representation (AMR), which helps to capture the nuances and diverse structures of utterances and understand connections between their varying levels of granularity. They evaluate the proposed method on two multi-domain NLU datasets (MTOP, MASSIVE) under both zero-shot and few-shot multi-domain settings. The results show that their method could improve LLMs’ capability in multi-grained NLU tasks.

**Questions For The Authors:**

1. Page 7, line 508. The paper says ”our proposed methods is LLM-agnostic”. But the experiments only involve gpt-3.5-turbo without comparing with other LLMs. Could you add the analysis of different LLMs?
2. Table 1. The experiments demonstrate different results of different prompts (zero/few-shot settings) without considering other related works. Could you add the results of other methods to make more comprehensive comparison? (e.g., full training. You can search the citations of MTOP[1], MASSIVE[2])

 [1] Li H, Arora A, Chen S, et al. MTOP: A comprehensive multilingual task-oriented semantic parsing benchmark[J]. arXiv preprint arXiv:2008.09335, 2020.
 [2] FitzGerald J, Hench C, Peris C, et al. Massive: A 1m-example multilingual natural language understanding dataset with 51 typologically-diverse languages[J]. arXiv preprint arXiv:2204.08582, 2022.

**Reasons To Accept:**

1. This work conducts the first preliminary study of LLMs’ capability in multi-grained NLU tasks, which is an interesting topic to be explored. It clearly describes the background (multi-grained NLU, LLMs, CoT and structured representation) before introducing the details of methods.
2. The method proposed in this paper is incremental, simple but reasonable, which decomposes the NLU tasks into multiple steps, and each step target at extracting certain information of the tasks.

**Reasons To Reject:**

1.	The weakness of this paper is that the experimental analysis is not strong enough. See "Questions For The Authors".
2.	A crucial question should be answered in the paper. What is the advantage of applying LLMs to NLU tasks? How about the comparison with the existing deep learning based NLU methods? As this is the first try to applying LLMs to multi-domain NLU tasks. Only when this question is answered, the proposed method makes sense.

**Reproducibility:**

3: Could reproduce the results with some difficulty. The settings of parameters are underspecified or subjectively determined; the training/evaluation data are not widely available.

**Reviewer Confidence:**

2: Willing to defend my evaluation, but it is fairly likely that I missed some details, didn't understand some central points, or can't be sure about the novelty of the work.

---

> ### Author Rebuttal · Authors · 2023-08-29
>
> Thank you for the feedback. We respond to your questions as follows:
>
> **(1) LLM-Agnostic Capability of CoF-CoT (Q1, RR1):**
> Our proposed method is LLM-agnostic since the focus of the work is on the prompt design, which can be applied to any LLMs. This is further validated via additional empirical studies presented in Table A. As most LLMs (i.e. GPT-3, PaLM, etc.) rely on the quality of the designed prompts, our proposed CoF-CoT prompt design can be used as input to any LLMs for zero-shot and in-context learning. This is similar to CoT, SC-CoT and other methods. We provide a detailed Prompt Design Template in Figure 3 (Appendix) for future extensions towards other LLMs.  For further clarification, we report additional results of our proposed method over the backbone PaLM LLM [1] on the MTOP dataset [2] below (Zero-shot Learning (ZSL) and Few-shot Learning (FSL) results are reported in the respective order from left to right). The format of the table below is similar to our main result Table 1 in the manuscript. We will update the newly reported results in the final version.
>
> |  | Intent Acc | Slot F1 |Frame Acc | Exact Match | Intent Acc | Slot F1 |Frame Acc | Exact Match |
> |-----------|-----------|-----------|-----------|-----------|-----------|-----------|-----------|----------|
> | PaLM  | 16.67 $\pm$ 2.52 | 7.24 $\pm$ 1.00 |3.17 $\pm$ 0.76 | 1.17 $\pm$ 0.76 | 48.83 $\pm$ 4.54 | 14.24 $\pm$ 1.58  | 4.17 $\pm$ 2.02  |2.33 $\pm$ 0.76 |
> | **PaLM + CoF-CoT** | **42.33 $\pm$ 3.33** | **13.73 $\pm$ 2.88** | **4.01 $\pm$ 0.15**  | **3.50 $\pm$ 1.32** | **57.17 $\pm$ 3.79** | **21.47 $\pm$ 3.79** | **10.33 $\pm$ 3.06** | **6.67 $\pm$ 2.75** |
>
> Table A: Performance of CoF-CoT integration with PaLM LLM on MTOP dataset.
>
> **(2) Comparison with fine-tuning (FT) approaches of Pre-trained Language Models (PLM) (Q2, RR2):**
> We argue traditional FT (i.e. full parameter training as proposed in [2],[3]) and LLM ZSL/FSL approaches are not directly comparable. Traditional FT operates under closed-world assumption which requires sufficient data to learn domain-specific and domain-agnostic feature extraction in multi-domain settings. Without sufficient data in  ZSL/FSL settings, FT approaches might not be able to capture these features, leading to inferior performance, especially when the domain gap between training and testing is significant. On the other hand, LLM provides an open-domain approach where fine-tuning on specific domains is not needed.
>
>
> We conducted a preliminary study and observed the aforementioned gap between the two approaches. For fair comparison between FT and LLM approaches, our study imposes the constraint that  there exist no overlapping domains between train and test domains under ZSL and FSL setting. We experimented with 3 different scenarios for FT approaches, including:
>
> * *Fully Supervised:* Samples sharing similar domains with test sets are used for training
> * *ZSL:* We use samples from domains different from test domains for training
> * *FSL:* We use samples from domains different from test domains in conjunction with a fixed number of  k-shot test domain samples.
>
> We leverage Roberta PLM [4] with joint Slot Filling and Intent Detection objectives as presented in [2],[3] as the FT model. Under ZSL and FSL settings, the FT model suffers from the aforementioned domain gap issues. Specifically, as observed in Table B, since there exist minimal overlapping intent labels between train and test domains, without sufficient data in ZSL and FSL settings, the FT approaches are unable to learn transferable multi-domain features, leading to 0 performance in Intent Accuracy. This behavior also results in 0 performance for both Frame Acc and Exact Match as the correct intents are the prerequisites for correct semantic frame and exact match metrics. On the other hand, Fully supervised FT approach acquires domain-specific knowledge of target domains from training data and performs the best across different evaluation metrics. However, this assumption does not match ZSL/FSL settings.
>
>
> |  Method | Assumption | Intent Acc | Slot F1 |Frame Acc | Exact Match |
> |-----------|-----------|-----------|-----------|-----------|-----------
> | Roberta FT  | Fully Supervised | 67.19 $\pm$ 2.90 | 75.17 $\pm$ 1.08 |43.57 $\pm$ 4.18 | 36.10 $\pm$ 1.08 |
> | Roberta FT | ZSL| 0| 12.68 $\pm$ 1.25 | 0 | 0|
> | Roberta FT | FSL| 0| 13.75 $\pm$ 1.22 | 0 | 0|
> | GPT3.5 + CoF-CoT | ZSL| 57.67 $\pm$ 2.75 | 23.47 $\pm$ 4.09 |  14.33 $\pm$ 1.52| 9.00 $\pm$ 1.00|
> | GPT3.5 + CoF-CoT | FSL| 61.50 $\pm$ 4.93 | 30.12 $\pm$ 3.93 |  15.00 $\pm$ 1.32| 11.00 $\pm$ 1.61|
>
> Table B: Complete comparison between FT and LLM approaches on MTOP dataset.
>
> **Reference:** RR1: Reason to Reject 1, Q1: Question 1
>
> [1] Chowdhery et al., Palm: Scaling language modeling with pathways. 2022. arXiv preprint arXiv:2204.02311.
>
> [2] Li et. al, MTOP: A Comprehensive Multilingual Task-Oriented Semantic Parsing Benchmark. EACL 2021.
>
> [3] FitzGerald et al., MASSIVE: A 1M-Example Multilingual Natural Language Understanding Dataset with 51 Typologically-Diverse Languages. 2022. arXiv preprint arXiv:2204.08582.
>
> [4] Liu et. al, RoBERTa: A Robustly Optimized BERT Pretraining Approach. 2019. arXiv:1907.11692

---

### Meta-Review · Area_Chair_ru3B · 2023-09-19

**Recommendation:** 4

**Metareview:**

The paper investigates the application of Chain-of-Thought (CoT) to Large Language Models (LLMs) in Natural Language Understanding (NLU). The proposed CoF-CoT approach enables LLMs to acquire and leverage essential concepts for solving tasks at different granularities. The authors also integrate structured knowledge via Abstract Meaning Representation (AMR) to capture the nuances and diverse structures of utterances. The paper evaluates the method on two multi-domain NLU datasets and shows improvement in LLMs' capability in multi-grained NLU tasks.

The reviewers generally find the topic and main contributions of the paper interesting. The proposed method is considered reasonable and incremental. However, there are concerns about the experimental analysis, lack of strong baselines, and comparisons with other methods. Reviewers also suggest exploring the advantages of different LLMs and providing more comprehensive comparisons with related works. Additionally, there are questions about implementing the proposed method in zero-shot and few-shot scenarios and its applicability when intermediate ground truths are not available.

Overall, the reviewers acknowledge the merits of the paper, such as reporting state-of-the-art results and presenting a novel idea. However, there are also key weaknesses, and the paper would benefit from another round of revision.

---

### Decision · Program_Chairs · 2023-10-07

**Decision:**

Accept-Main

**Comment:**

The paper investigates the application of Chain-of-Thought (CoT) to Large Language Models (LLMs) in Natural Language Understanding (NLU). The proposed CoF-CoT approach enables LLMs to acquire and leverage essential concepts for solving tasks at different granularities. The authors also integrate structured knowledge via Abstract Meaning Representation (AMR) to capture the nuances and diverse structures of utterances. The paper evaluates the method on two multi-domain NLU datasets and shows improvement in LLMs' capability in multi-grained NLU tasks.

The reviewers generally find the topic and main contributions of the paper interesting. The proposed method is considered reasonable and incremental. However, there are concerns about the experimental analysis, lack of strong baselines, and comparisons with other methods. Reviewers also suggest exploring the advantages of different LLMs and providing more comprehensive comparisons with related works. Additionally, there are questions about implementing the proposed method in zero-shot and few-shot scenarios and its applicability when intermediate ground truths are not available.

Overall, the reviewers acknowledge the merits of the paper, such as reporting state-of-the-art results and presenting a novel idea. However, there are also key weaknesses, and the paper would benefit from another round of revision.